# Calculation Model of Multi-Well Siphoning and Its Feasibility Analysis of Discharging the Groundwater in Soft Soil

Qingsong Shen [1], Chaofeng Wu [2], Jun Wang [3], Shuai Yuan [3], Yuequan Shang [1] and Hongyue Sun [3,*]

[1] Department of Civil Engineering, Zhejiang University, Hangzhou 310058, China
[2] China Energy Engineering Group Zhejiang Electric Power Design Institute, Hangzhou 310012, China
[3] Ocean College, Zhejiang University, Zhoushan 316021, China
* Correspondence: shy@zju.edu.cn

**Abstract:** The use of a siphon is a drainage method that does not require the use of external power, and siphons are used extensively in practice. The application of siphons in the treatment of soft soils has become a popular approach in recent years. Analyzing the change in the water level in multi-well siphoning is the basis of the application of siphons. The calculation model of multi-well siphoning is established through equivalent transformation. A finite difference method is used to solve the model, and the accuracy is verified by comparing the results with a test of the model and with field tests. Using the numerical model, the feasibility of siphoning drainage in soft soil is demonstrated from two aspects, i.e., well spacing and the permeability coefficient. The results show both the "minimum drop increase" and the "average drop increase", as well as the spacing of the decreases or increases in the permeability coefficient. When the permeability coefficient is $1 \times 10^{-8}$ m/s and the well spacing is 2 m, the decrease in the water level is approximately 9.72 m after 100 days of drainage. It is feasible to apply siphon drainage technology to discharge the groundwater in soft soil.

**Keywords:** siphon drainage; discharge groundwater; fixed-drop; foundation treatment





## 1. Introduction

Soft soil is widely distributed throughout the world. It is characterized by high compressibility, large water content, poor permeability, and low strength [1–3], and it always has been a complex problem in research related to geotechnical engineering. Soft soils typically are treated through the drainage consolidation method, which usually consists of a drainage system and a loading system [4,5]. The drainage system is composed of vertical drainage wells and horizontal drainage cushions. The loading system applies a load pressure to the soil and causes the water in the pores to seep out, resulting in the consolidation of the soil. There are two common methods for preloading, i.e., surcharge preloading [6] and vacuum preloading [7,8]. However, the former requires heavy loads, and the process is time-consuming and labor-intensive [9]. The latter requires the air pump to be operating continuously to facilitate vacuum pumping. In addition, there will be clogging problems that affect the consolidation during the vacuum preloading process [10].

The siphon effect is created by molecular gravity and potential energy difference, and it allows liquids to be transported without the use of any external power [11–13]. This method of drainage is environmentally friendly, economically feasible, convenient, and effective. In previous research, siphon drainage technology has been applied effectively to deal with potential landslides caused by rainfall [14–16]. However, the application of this technology in the consolidation of soft soil is still in the experimental stage [17,18].

Although siphon drainage and well-point dewatering have some similarities, they also have some differences. Figure 1 shows a schematic diagram of siphon drainage in which drainage wells are arranged in soft soil. The inlets of the siphon pipes are placed in the drainage wells, and the outlets are placed in a collection well. A pump is used to

maintain the water level in the collection well below the siphon well. After siphon drainage begins, no external power is required for the groundwater to flow through the siphon pipes into the collection well. In response to the decrease in the level of the water, the effective stress of the soil increases, and this results in the consolidation of the soil. In addition, it creates the conditions required to implement other ground treatment methods. The siphons that are placed throughout the entire treatment area drain all of the water, and no electricity is used within the area. In comparison to conventional well-point dewatering, the process described above saves more energy and is more environmentally friendly. The success of soft soil siphon drainage technology depends on whether or not it can reduce groundwater levels.

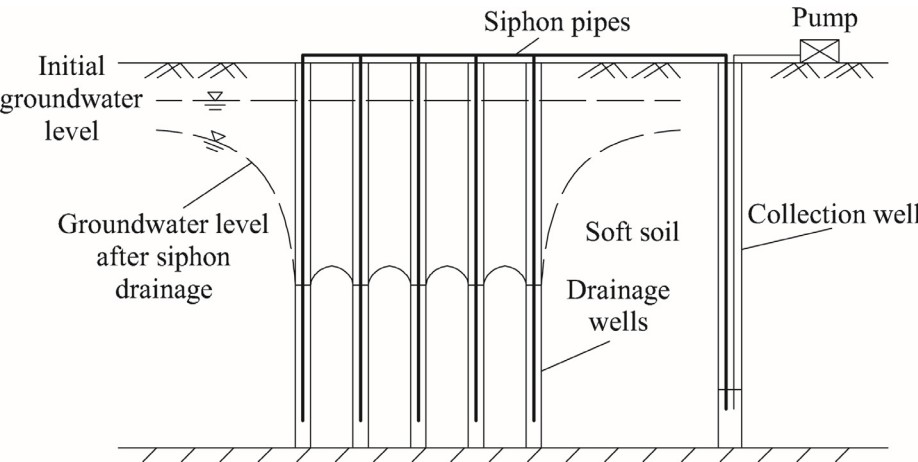

**Figure 1.** Schematic diagram of siphon drainage in soft ground.

Researchers have conducted a considerable amount of theoretical work on the effects of pumping on water level. A formula for single-well pumping at a constant discharge rate in confined aquifers was proposed by Theis [19], and it provides the basis for the theory of unsteady well flow. Hantush et al. [20] provided the well flow formulas for single-well pumping of partially penetrated wells in leaky aquifers. Neuman [21] separately calculated the well flow formulas for single-well pumping of fully penetrated and partially penetrated wells in phreatic aquifers. Sen [22] considered the influence of well diameter and the well storage effect of pumping wells and obtained the corresponding unsteady well flow solution. Cohen and Rabold [23] considered the non-uniform flow of the pumping well wall and obtained the semi-analytical solution of incomplete well flow by integrating at the boundary.

The above theories are based on the assumption that the aquifer extends infinitely in the horizontal direction, and they discuss the changes in the water level that are caused by the pumping of a single well. However, the majority of practical projects involve the use of multiple drainage wells. Due to the interaction between different drainage wells, the rate at which the water level decreases among multiple wells will be greater than the rate at which the water level decreases in a single well. Thus, the single-well drainage theory with infinite lateral extension is not applicable to the drainage of multiple wells. A common assumption made in the theory of multi-well drainage is that the discharge rate of the drainage wells is constant, and the superposition principle is used in order to determine the water level [24,25]. However, multi-well siphoning is characterized by intermittent drainage rather than a fixed drainage rate. Due to the water catchment effect, the siphon will begin to operate once the water level in the well reaches the starting height. In the course of the drainage process, the water level will decrease, resulting in the interruption of the siphon. These two processes will be repeated continuously by the siphon drainage. Considering the fact that the drainage rate will change over time, the existing theories cannot be applied effectively to the complex problem of multi-well siphoning. In this paper, the problem of multi-well siphoning in homogeneous and isotropic soils is considered.

According to the principle of symmetry, wells are divided into multiple identical single wells with boundary conditions and constraints. The water level in multi-well siphoning can be calculated. The flowchart of the methodology is shown in Figure 2.

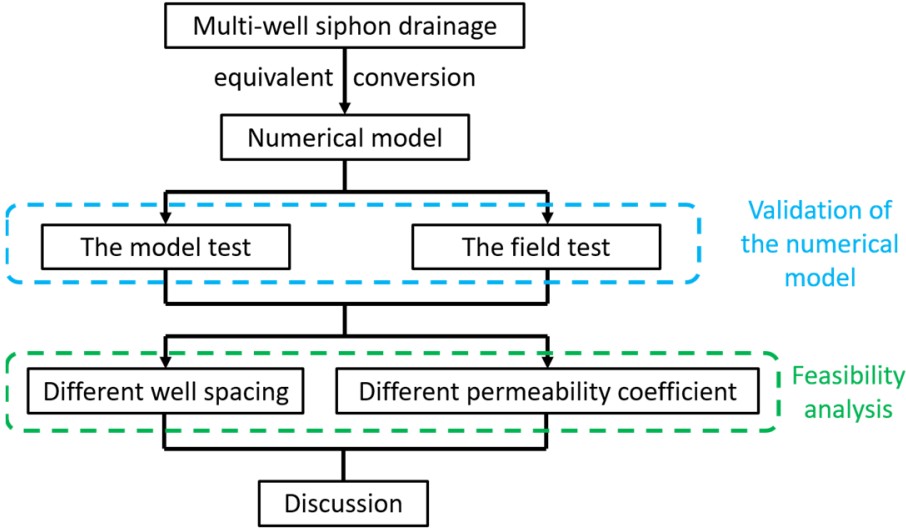

**Figure 2.** Flowchart of the methodology.

## 2. Method

### 2.1. Equivalent Conversion

Wu et al. [18] considered that the water level in the drainage wells would be kept at the design height. Accordingly, siphon drainage can be considered to be a problem of fixed-drop well flow. In addition, the experiment showed that, for siphon pipes that have 4 mm diameters, the starting height of the siphon is 0.13 m when the siphon lift is 9.75 m. The starting height amounts to only 1.3% of the siphon lift, which may not be considered important in engineering applications. Therefore, it is a reasonable and practical measure to consider siphon drainage as a fixed-drop well flow problem.

There is a specific sequence in which siphon drainage wells are set in the practical project. Figure 3a,b are two common layout forms, i.e., the equilateral triangle and the square. Assuming that the soft soil layer is homogeneous and isotropic, the decrease in the water level of each siphon drainage well is the same. According to the symmetry principle, the seepage field between the two drainage wells is symmetrically distributed about the axis of symmetry. At the position of the axis of symmetry, there is no seepage movement in the horizontal direction, and the slope of the tangent line to the water level line at this point is zero. Hence, as shown in Figure 4, the wells can be divided into multiple identical wells. Figure 3 shows that the division method is a hexagon when the drainage wells are arranged in an equilateral triangle, and it is a square when they are arranged in a square. When the numerical model is obtained after conversion, it can be modelled as a circle to solve the problem, and the relationship between the equal radius, $r_e$, and the well spacing, $l$, is as follows [26,27]:

When set in an equilateral triangle:

$$r_e = \frac{1}{2}\sqrt{\frac{2\sqrt{3}}{\pi}l} = 0.525l \tag{1}$$

When set in a square:

$$r_e = \frac{1}{2}\sqrt{\frac{4}{\pi}l} = 0.564l \tag{2}$$

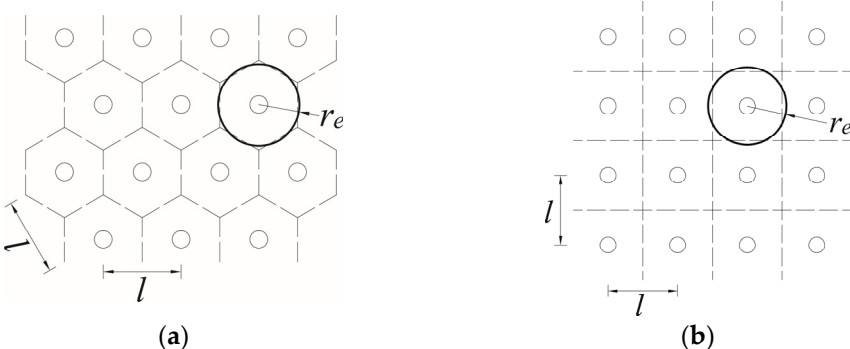

**Figure 3.** Division method under different drainage well layouts: (**a**) equilateral triangle, (**b**) square.

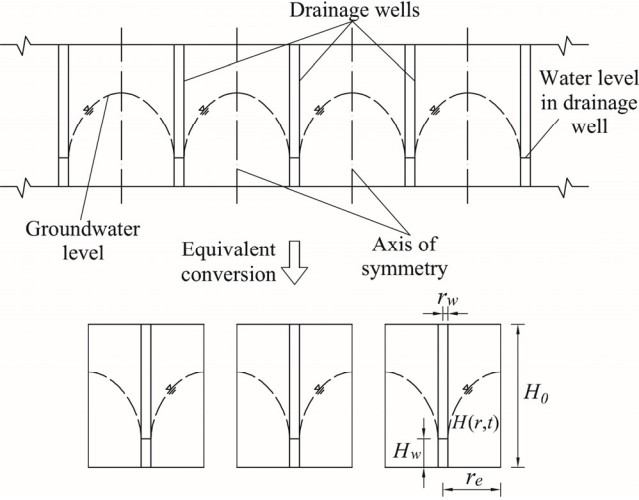

**Figure 4.** Schematic diagram of groundwater level division of multi-well siphoning.

*2.2. Solution of the Numerical Model*

Figure 3 shows the cross-sectional schematic diagram of the numerical model. The thickness of the aquifer is $H_0$, and the equivalent radius is $r_e$. The initial water head is at the top of the aquifer. That is, the initial groundwater level is $H_0$. The radius of the drainage well is $r_w$. The water level of the drainage well is fixed at $H_w$. The bottom of the model is an impervious boundary. The soft clay layer is regarded as a phreatic aquifer. The following basic assumptions are made: (1) The phreatic aquifer is homogeneous, isotropic, uniformly thick, and horizontally distributed. (2) The seepage meets Darcy's law and Dupuit's assumption. (3) The cross-flow phenomenon, external infiltration, and water evaporation are not considered. (4) The well is fully penetrated, and the water level of the drainage well is fixed. (5) The water flow at the equivalent radius boundary does not allow seepage in the horizontal direction. (6) The groundwater release caused by the decrease of the water head is completed instantaneously. Based on the above assumptions, an equation of spatial, two-dimensional, axisymmetric groundwater seepage movement can be established [19] as follows:

$$a\left(\frac{\partial^2 \varphi}{\partial r^2} + \frac{1}{r}\frac{\partial \varphi}{\partial r}\right) = \frac{\partial \varphi}{\partial t}, (r_w < r < r_e, t > 0) \tag{3}$$

where $a$ is the hydraulic coefficient. For the flow of the phreatic aquifer $a = Kh_m/\mu$, $K$ is the permeability coefficient, $h_m$ is the average thickness of the aquifer, $\mu$ is the water supplied by gravity, and $\varphi$ is the well flow potential function of the phreatic aquifer, $\varphi = H^2/2$.

The initial conditions of the numerical model can be described as:

$$\varphi(r,0) = \frac{1}{2}H_0{}^2 \tag{4}$$

The boundary conditions of the numerical model at any time can be described as:

$$\varphi(r_w, t) = \frac{1}{2}H_w{}^2 \tag{5}$$

$$\frac{\partial}{\partial r}\varphi(r_e, t) = 0 \tag{6}$$

The finite difference method is used to solve the equation. The explicit difference format is constructed as follows:

$$\frac{\partial \varphi}{\partial r} = \frac{\varphi_{j+1}^i - \varphi_j^i}{dr} \tag{7}$$

$$\frac{\partial^2 \varphi}{\partial r^2} = \frac{\frac{\varphi_{j+1}^i - \varphi_j^i}{dr} - \frac{\varphi_j^i - \varphi_{j-1}^i}{dr}}{dr} = \frac{\varphi_{j+1}^i - 2\varphi_j^i + \varphi_{j-1}^i}{(dr)^2} \tag{8}$$

$$\frac{\partial \varphi}{\partial t} = \frac{\varphi_j^{i+1} - \varphi_j^i}{dt} \tag{9}$$

Substituting Equations (6)–(8) into Equation (1) allows the formula of the discrete internal points to be calculated:

$$\varphi_j^{i+1} = \varphi_j^i + a \cdot \left[ \frac{\varphi_{j+1}^i - 2\varphi_j^i + \varphi_{j-1}^i}{(dr)^2} + \frac{1}{r} \cdot \frac{\varphi_{j+1}^i - \varphi_j^i}{dr} \right] \cdot dt \tag{10}$$

Equation (10) indicates that the well flow potential function at the time $i + 1$ can be calculated from the three nodes, i.e., $\varphi_{j+1}^i$, $\varphi_j^i$, and $\varphi_{j-1}^i$. According to Equations (4) and (5), the calculation formula of the discrete points on the boundary can be obtained as follows:

$$\varphi_j^1 = H_0, \varphi_1^i = H_w \tag{11}$$

The discrete points $\varphi_{N-1}^i$ and $\varphi_{N-2}^i$ can be expanded at $\varphi_N^i$ by using Taylor's formula:

$$\varphi_{N-1}^i = \varphi_N^i + \varphi_N^i{}'(-dr) + \varphi_N^i{}''(-dr)^2 + o(-dr) \tag{12}$$

$$\varphi_{N-2}^i = \varphi_N^i + \varphi_N^i{}'(-2dr) + \varphi_N^i{}''(-2dr)^2 + o(-2dr) \tag{13}$$

Ignoring the high-order infinitesimal term and substituting Equation (6) into Equations (12) and (13), the calculation formula of the discrete point $\varphi_N^i$ can be obtained:

$$\varphi_N^i = \frac{4\varphi_{N-1}^i - \varphi_{N-2}^i}{3} \tag{14}$$

Equations (10), (11) and (14) are the calculation formulas for solving all discrete points in the solution domain. The calculation results of the numerical model can be obtained by using MATLAB to write the corresponding program.

## 3. Validation of the Numerical Model

### 3.1. The Model Test

According to the symmetry principle, the problem of multi-well siphoning is solved by dividing wells into multiple, identical, single wells. Through Equations (1) and (2), it can be modelled as a circle. Therefore, a cylindrical physical model test is carried out to verify the accuracy of the numerical model. Such cylindrical tests are very common in the groundwater field [28]. In the rectangular model, the water level is related to three

parameters, i.e., x, y, and z. In the cylindrical model, the water level is related to θ and z. The cylindrical model can reduce the number of parameters and make calculation easier.

The test uses an acrylic cylinder. The height is 90 cm, the inner diameter is 32 cm, and the outer diameter is 34 cm. The thickness is 1 cm, which prevents excessive lateral deformation during the test. The soil sample used is clay from a construction site in Zhoushan. The size of the PVD is 10 cm × 0.4 cm, which equals $(a + b)/\pi = 3.3$ cm. The siphon pipes are PU pipes with an inner diameter of 4 mm. The piezometer pipes are PU pipes with an inner diameter of 6.5 mm. Figure 5 shows the physical model test.

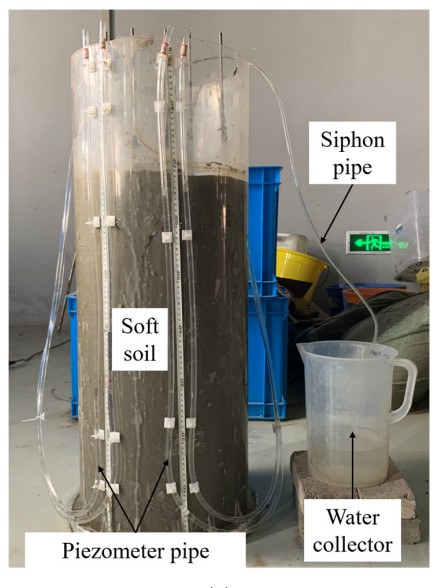

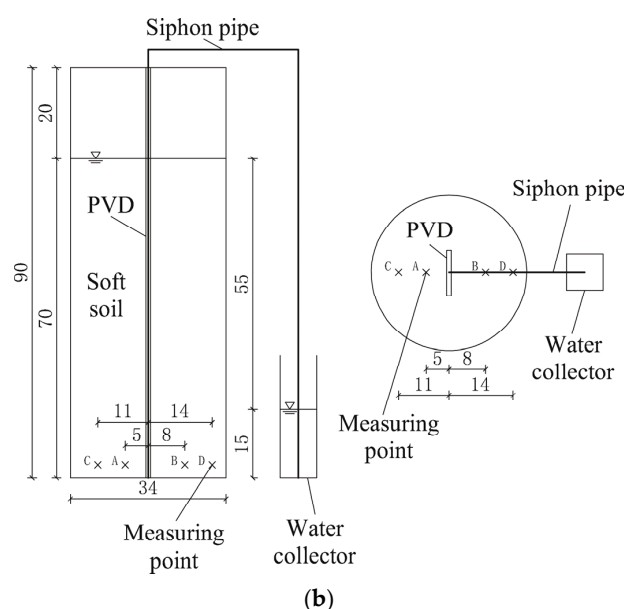

(a)                                                                              (b)

**Figure 5.** The model test: (**a**) photographic image,(**b**) schematic diagram.

Wrap a geotextile at one end of the siphon pipe. The geotextile material is woven geotextile and weighs 100 g per square meter. The length of the package is 3 cm, and the number of layers is 2. As a result of this step, soil particles are prevented from entering the siphon pipe and clogging the pipe. The siphon pipe is inserted into the gap between the plastic core board and the filter membrane. The PVD with the siphoning effect is completed. Drill holes evenly on the surface of the PVC pipe and insert the piezometric pipe into the holes. After the piezometer pipe is fixed and the geotextile is wrapped evenly around the PVC pipe, the water level measuring device is completed. According to the schematic diagram, the PVDs are set at the center of the cylinder. The water level measuring devices are arranged at 5, 8, 11, and 14 cm away from the PVD. The soil sample retrieved from the site is soaked in water for one week, after which the soil sample is beaten with a beater until it becomes silt or mud without agglomeration. The particle size distribution of the soil sample is tested by the laser particle size distribution instrument BT-9300Z. The results for this study are shown in Figure 6. The geotechnical test showed that the water content of the soil sample was 53%, the specific gravity of the particles was 2.71, and the void ratio was 1.44. The sample is filled into the cylinder to a height of 70 cm. After standing for more than three days, the sample is consolidated under its weight. The water level in the water collector is kept stable at 15 cm from the ground. Under the action of the height difference of the liquid level, the siphon pipe starts to drain. Record the water level of each measuring point.

Since the permeability coefficient of the soil sample is unknown, a permeability test is still required. To keep the size consistent with the original test, the device used in the permeability test is still the cylinder. The bottom of the cylinder should be covered with a sand cushion that is 10 cm thick. The geotextile is laid on the sand cushion to prevent soil particles from entering the sand cushion. The sample is filled into the cylinder to the height

of 20 cm. The water level in the cylinder should be maintained at a constant position. Record the flow at the outlet. After three sets of tests, the calculated permeability coefficients in this study were $8.1 \times 10^{-4}$, $7.9 \times 10^{-4}$, and $7.8 \times 10^{-4}$ m/d. It is estimated that the average value of the permeability coefficient of the soil was approximately $7.93 \times 10^{-4}$ m/d.

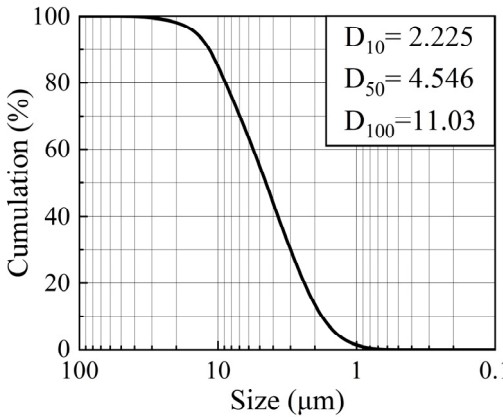

**Figure 6.** Particle size distribution of the soil sample.

The parameters used in the calculation of the numerical model are shown in Table 1. The comparison of the results of the numerical solution and the model test is shown in Figure 7. The curve represents the numerical model, and the dots represent the model test. The numerical solution of the decrease in the water level at C and D agrees with the test value. The numerical solution of the decrease in the water level at A deviated slightly from the test value. This occurred because point A was located closer to the drainage well, and the rate of change in the water level was relatively high. Figure 7a indicates that the water level decreased at a greater rate when the distance from the drainage well was smaller. On the contrary, when farther from the drainage well, the water level dropped at a smaller rate. The groundwater level line presented an apparent "funnel shape". Figure 7b shows that the decrease in the water level at each position gradually increased and then tended to remain flat. According to the above analysis, the numerical solution and test value were in good agreement, and the trend between them was consistent. Figure 8 shows the relative error between the model test and the numerical model. Except for the initial stage of drainage (days 0–3), the relative error in the whole drainage process was basically within 10%. Especially after the 10th day, the relative error had been maintained within 5%. Moreover, the relative error at the initial stage of drainage was only slightly more than 10%, i.e., still less than 15%. As a result, it has been verified that the siphon drainage model was correct.

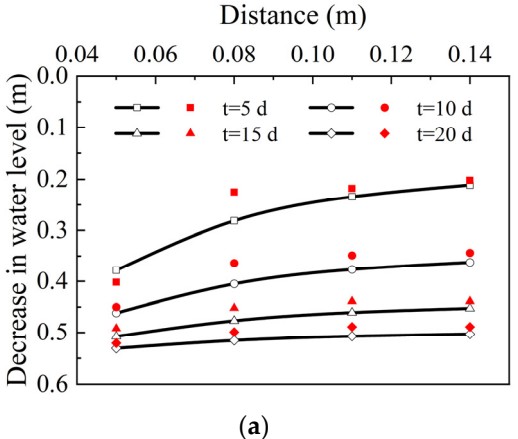

(**a**)

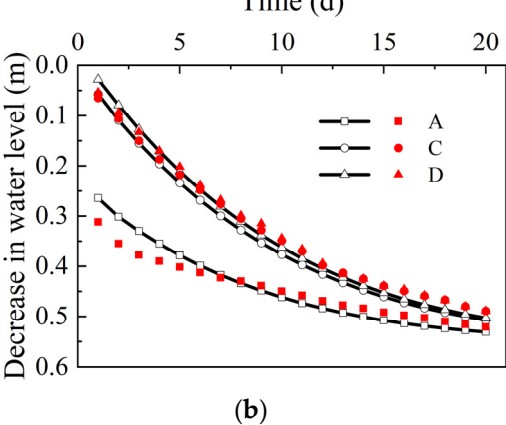

(**b**)

**Figure 7.** Comparison of the numerical model (curves) and the model test (dots): (**a**) variation of the decrease in the water level with distance, (**b**) variation of the decrease in the water level with time.

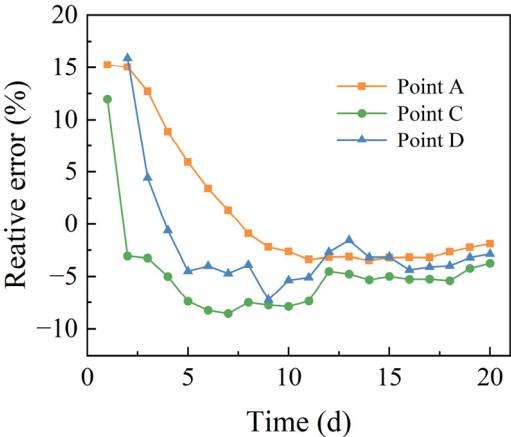

**Figure 8.** The relative error between the numerical model and the model test.

**Table 1.** Parameters used in the calculation of the numerical model.

| $r_e$ | $r_w$ | $H_0$ | $H_w$ | $K$ | $\mu$ | $a$ | $dr$ | $dt$ |
|---|---|---|---|---|---|---|---|---|
| 0.16 m | 0.033 m | 0.7 m | 0.15 m | $7.93 \times 10^{-4}$ m/d | 0.2 | $2.8 \times 10^{-3}$ m$^2$/d | 0.01 m | $1 \times 10^{-3}$ d |

*3.2. The Field Test*

The numerical model was compared with the groundwater level data obtained from the field test of multi-well siphon drainage [29].

The field test of multi-well siphon drainage was located in the northern reclamation area of Dapeng Island, Zhoushan City. Initially, hydraulic reclamation technology was used in the area to form a land area. Table 2 shows the geological conditions of the soil layers at the site. Table 2 shows that the soil in the region has high water content, a large void ratio, high compressibility, and low permeability.

**Table 2.** Geological conditions of the soil layers at the site.

| Layers | Dredger Soil | Mucky Clay | Mucky Clay |
|---|---|---|---|
| Thickness | 10.2 m | 6.1 m | 9.4 m |
| Water content | 47.2% | 35.8% | 42.4% |
| Specific gravity | 2.74 | 2.72 | 2.74 |
| Void ratio | 1.318 | 1.052 | 1.204 |
| Liquid limit | 40.2% | 34.5% | 40.2% |
| Plastic limit | 22.2% | 20.6% | 22.2% |
| Horizontal permeability coefficient | $5.32 \times 10^{-9}$ m/s | $6.24 \times 10^{-9}$ m/s | $3.97 \times 10^{-9}$ m/s |
| Vertical permeability coefficient | $6.95 \times 10^{-9}$ m/s | $8.12 \times 10^{-9}$ m/s | $5.10 \times 10^{-9}$ m/s |
| Compressive modulus | 2.89 MPa | 2.57 MPa | 2.42 MPa |

The distribution of siphon drainage holes in the field test is shown in Figure 9. The siphon drainage wells are arranged in a square configuration, with a total of 18 rows and 13 columns of siphon drainage wells. In the 4th through the 18th rows, the well spacing is 0.9 m. According to Equation (2), the equivalent radius of the numerical model is 0.51 m. The water level in the siphon drainage wells is maintained at 10 m from the surface. The radius of the siphon drainage wells is 0.04 m, and the average depth of the siphon drainage wells is 18 m. Select the soil within the depth range of the siphon drainage well to calculate the water level. The overall permeability coefficient of soil is the average value of the dredger soil, i.e., $6.14 \times 10^{-9}$ m/s. The parameters used in the calculation of the numerical model are shown in Table 3.

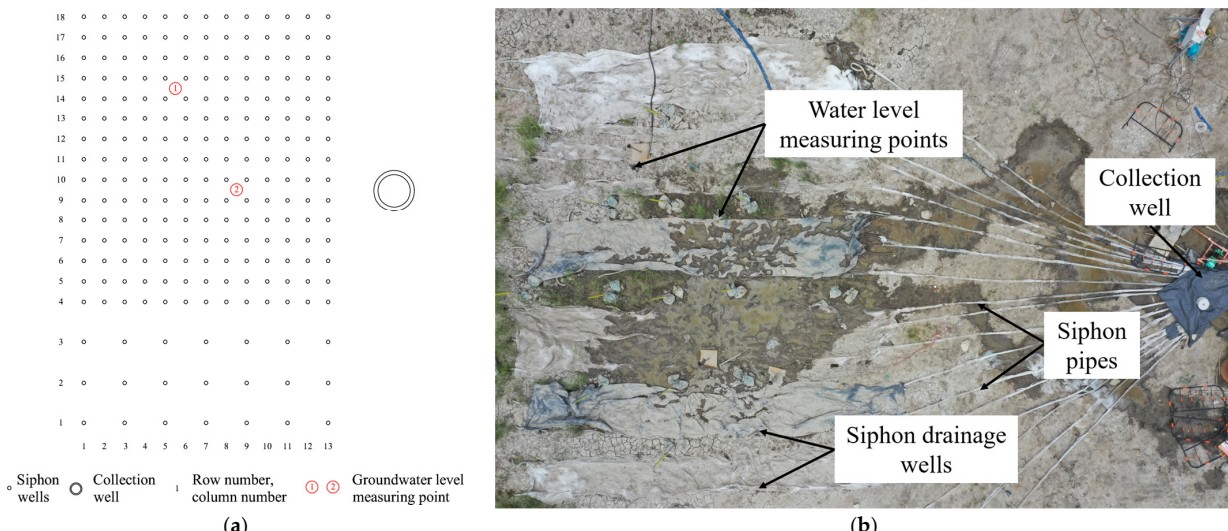

**Figure 9.** Field test of multi-well siphon drainage: (**a**) schematic diagram, (**b**) field diagram.

**Table 3.** Parameters used in the calculation of the numerical model.

| $r_e$ | $r_w$ | $H_0$ | $H_w$ | $K$ | $\mu$ | $a$ | $dr$ | $dt$ |
|---|---|---|---|---|---|---|---|---|
| 0.51 m | 0.04 m | 18 m | 8 m | $5.31 \times 10^{-3}$ m/d | 0.2 | $4.78 \times 10^{-2}$ m$^2$/d | 0.01 m | $1 \times 10^{-4}$ d |

The result of the comparison of the numerical solution and the field test is shown in Figure 10. The dots are the measured values of the field test, and the curve is the numerical solution of the model. The relative error between the result of the numerical model and the field test is shown in Figure 11. It can be seen in Figure 10 that the variations of the decrease in the water level obtained by the numerical solution and the field test were basically the same. With the increase in time, the water level drop increased gradually. When the drainage time was 16 days, the decrease in the water level at measuring points 1 and 2 increased to 2.39 m and 2.78 m, respectively. The decrease in the water level of the numerical model increased to 2.51 m. Similar to the results of the model test, there is a certain deviation between the numerical model and the field test at the initial stage of drainage. Subsequently, the two were in good agreement, and the relative error was basically less than 10%. The maximum relative error between the numerical model and groundwater level measuring point #1 was 10.1%, and the maximum relative error between the result of the numerical model and groundwater level measuring point 2 was 10.1%. Therefore, it can be shown that the numerical model has certain accuracy and provides a good description of the change in the water level during siphon drainage.

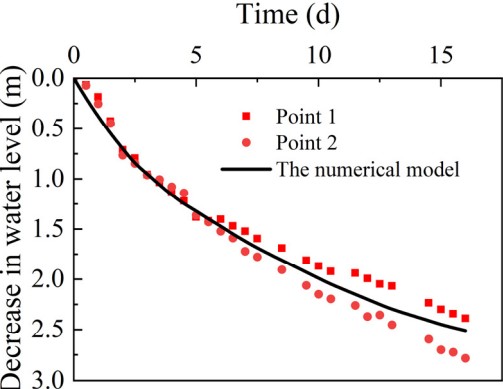

**Figure 10.** Comparison of the numerical model (curve) and the field test (dots).

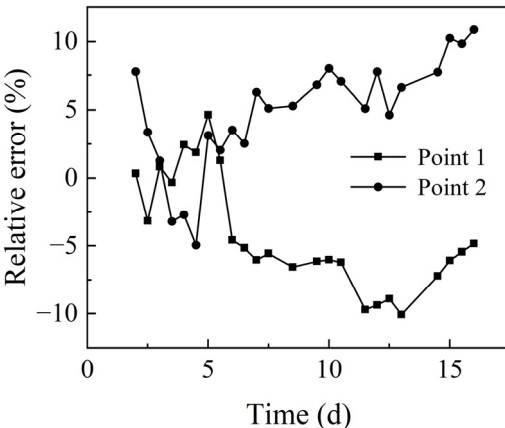

**Figure 11.** Relative error between the numerical model and the field test.

## 4. Feasibility Analysis

Soft soil siphoning utilizes siphon drainage, which lowers the water level, reduces the pore water pressure, and increases the effective stress of the soil, thereby causing it to compress and consolidate. Consequently, the key to the successful application of the siphon technology is the ability to reduce water in the soil. In the process of siphon drainage, the water level presents an apparent "funnel shape" along the radial direction. The water level drop at the midpoint between two adjacent wells is called the "minimum drop", which characterizes the maximum water level. The average value of the decrease in the level of the groundwater at all locations is called the "average drop", and it represents the overall water level in the drainage area. The effect of well spacing and permeability coefficients on the level of the water is analyzed based on the rates of change of these two parameters.

In actual projects, drainage wells will be arranged by different well spacings. It is assumed that the thickness of the phreatic aquifer is 15 m. The soil permeability coefficient is $1 \times 10^{-8}$ m/s. The radius of the drainage well is 0.03 m. The water level in the well is fixed at 10 m. The model is calculated with different well spacings. The changes in "minimum drop" (solid line) and "average drop" (dotted line) are shown in Figure 12. Figure 12a shows that both "minimum drop" and "average drop" increased gradually with the increase of time. Initially, the rate of change was fast, but it gradually became slower and then remained stable. Figure 12b shows that both "minimum drop" and "average drop" increased gradually as the well spacing decreased. A smaller well spacing simultaneously results in a greater drop in the level of the groundwater. It can be analyzed from the difference between "minimum drop" and "average drop" that the difference decreased gradually as the well spacing decreased. It is evident that when the well spacing was smaller, the "falling funnel" formed in the drainage process was smoother, resulting in a significant decrease in the level of the groundwater at all locations in the soil. For analysis, take the decrease in the level of the groundwater following a period of 100 days of drainage. When the well spacing was 8 m, the "minimum drop" was 0.89 m, and the "average drop" was 1.91 m. When the well spacing was reduced to 4 m, the "minimum drop" was 4.44 m, and the "average drop" was 5.21 m. When the well spacing was further reduced to 2 m, the "minimum drop" increased to 9.72 m, and the "average drop" increased to 9.77 m. It was about twice as big at 4 m and about nine times as big at 8 m, i.e., approximately equal to the limit drop. This shows that reducing the well spacing effectively can reduce the groundwater level in the soil.

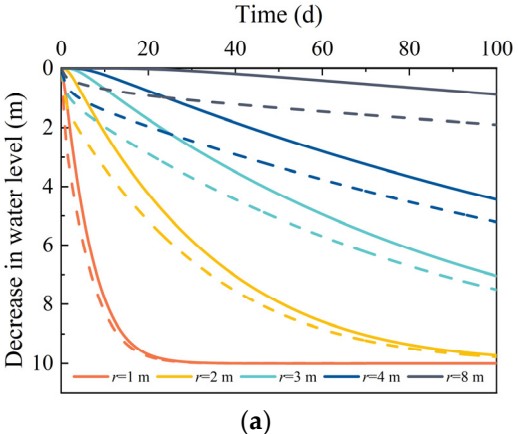 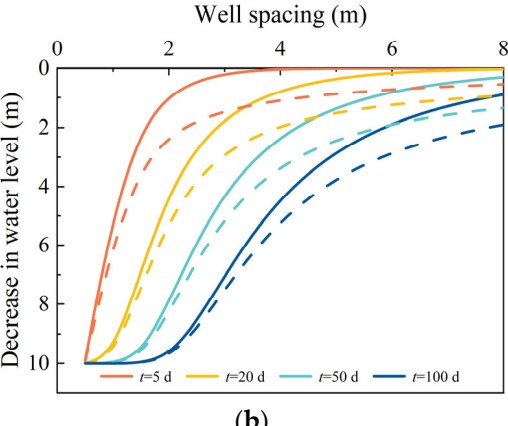

**(a)**　　　　　　　　　　　　　　　**(b)**

**Figure 12.** Variation of "minimum drop" (solid line) and "average drop" (dotted line) under different well spacings: (**a**) variation of the decrease of the water level with time, (**b**) variation of the decrease of the water level with well spacing.

The magnitude of the permeability coefficient usually is in the range of $10^{-9}$–$10^{-7}$ m/s. The well spacing is assumed to be 2 m, and the other parameters are unchanged. The model is calculated with different permeability coefficients within the value range. The changes in "minimum drop" (solid line) and "average drop" (dotted line) are shown in Figure 13. It can be seen from Figure 13a that both "minimum drop" and "average drop" gradually increase with the increase in time. Initially, the rate of change was fast, but it gradually became slower and then remained stable. This shows that, after a certain period of siphon drainage, the groundwater level can be reduced to the control height for soils with relatively high permeability. Although it ultimately cannot reduce the groundwater level of low permeability soils to the limit height, it still decreases this level to a certain extent. Figure 13b shows that, in the initial stage of drainage, both "minimum drop" and "average drop" increased uniformly with the increase in the permeability coefficient. In the late stage of the drainage, the two decreases in the groundwater level increased rapidly as the permeability coefficient increased, and then they stabilized at the limit drop value. This indicates that the greater the permeability coefficient becomes, the greater the decrease in the water level will be. However, due to the limitation of the siphon, the groundwater level will remain unchanged after it drops to the limited value. Figure 9 shows that the "minimum drop" and the "average drop" gradually approached each other in size as the permeability coefficient increased. This indicates that the greater the permeability coefficient, the more the groundwater level will decrease at all locations in the soil. Consider the decrease in the groundwater level for analysis with 100 days of drainage time. When the permeability coefficient was $6 \times 10^{-9}$ m/s, the "minimum drop" was 8.58 m, and the "average drop" was 8.84 m. Obviously, the groundwater level had decreased. When the permeability coefficient was $2 \times 10^{-8}$ m/s, both the "minimum drop" and the "average drop" were 10 m, which had reached the limit value. This shows that the siphon technology can provide a good drainage effect in soft soils with different permeability coefficients.

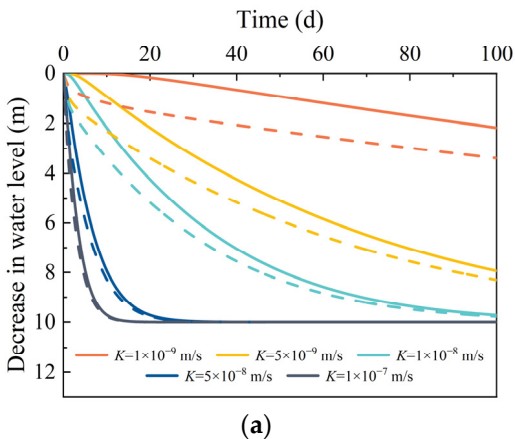
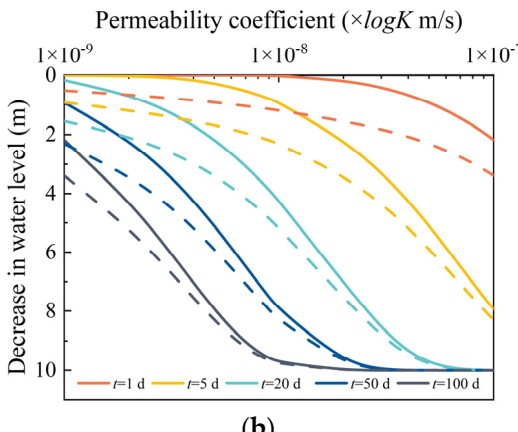

**(a)**                                                      **(b)**

**Figure 13.** Variation of "minimum drop" (solid line) and "average drop" (dotted line) under different permeability coefficients: (**a**) variation of water level drop with time, (**b**) variation of water level drop with permeability coefficient.

## 5. Discussion

From the previous analysis, it is apparent that the problem of siphon drainage well flow can be regarded as the problem of fixed-drop well flow. Jacob [30] assumed that the aquifer extended infinitely laterally, and the Laplace transformation was used to obtain the analytical solution of the fixed-drop well flow:

$$H = \sqrt{H_0^2 - \left(H_0^2 - H_w^2\right) A\left(\bar{r}, \bar{t}\right)} \tag{15}$$

$$A\left(\bar{r}, \bar{t}\right) = 1 - \frac{2}{\pi} \int_0^\infty \frac{J_0(z)Y_0(z\bar{r}) - Y_0(z)J_0(z\bar{r})}{z\left[J_0^2(z) - Y_0^2(z)\right]} e^{-\bar{t}z^2} dz \tag{16}$$

In the formula: $A\left(\bar{r}, \bar{t}\right)$ refers to the water level drop function of a fixed drop without an overflow system; $\bar{r} = r/r_w$ is the dimensionless radial distance; $\bar{t} = at/r_w^2$ is dimensionless time; $J_0(x)$ is a Bessel function of the first kind; $Y_0(x)$ is a Bessel function of the second kind, of zero order.

Assume that the same parameters in the numerical model and the Jacob model are shown in Table 4. The space step is 0.01 m and the time step is $1 \times 10^{-3}$ d. The equivalent radius of the numerical model is assumed to be 1.05 m. The water level drawdown at 0.35 m from the center of the drainage hole is selected for comparative analysis (Figure 14). It can be seen from the figure that when the drainage time is 0 to 30 days, the numerical model and the Jacob model basically provide identical results. When the drainage time exceeds 30 days, the deviation between the numerical model and the Jacob model begins to appear. The difference gradually increases with the increase in time. When the drainage time is 100 days, the decrease in the water level of the numerical model is 3.98 m, and it is 3.09 m in the Jacob model. Compared with the Jacob model, the numerical model shows an increase of 28.8%. This is because the Jacob model considers single-well drainage, and the numerical model in this paper considers multi-well drainage. At the initial stage of drainage, the impact of multi-well drainage is small, and there is no significant difference between single-well drainage and multi-well drainage. However, with the progress of drainage, the impact of multi-well drainage increases, and the increase of the drop of the water level of multi-well drainage is greater than that of single-well drainage. When the drainage duration is 100 days, the numerical model increases by 28.8% compared with the Jacob model.

**Table 4.** Parameters used in the calculation of the numerical model and the Jacob model.

| $r_w$ | $H_0$ | $H_w$ | $K$ | $\mu$ | $a$ |
|---|---|---|---|---|---|
| 0.05 m | 15 m | 5 m | $8.64 \times 10^{-4}$ m/d | 0.2 | $6.48 \times 10^{-3}$ m²/d |

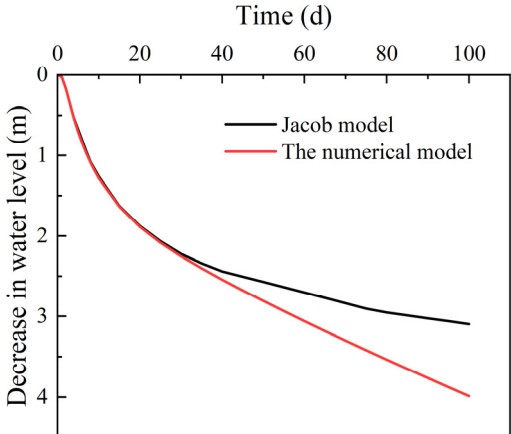

**Figure 14.** Comparison of the decrease in the water level between the numerical model and the Jacob model.

## 6. Conclusions

The calculation model of multi-well siphoning was established, and a finite difference method was used to solve the model. The accuracy of the model was verified by comparing the results with a test of the model and with field tests. Using the numerical model, the feasibility of siphoning drainage in soft soil was demonstrated. The conclusions of the research can be summarized as following:

(1) When the soil is homogeneous and isotropic, the seepage field between the two drainage wells is distributed symmetrically. The wells can be divided into identical single wells so that the water level change in multi-well siphoning can be calculated. Comparing the numerical model with the model test and the field test, the relative error was within 10%.

(2) It is feasible to apply siphon drainage technology to discharge the groundwater in soft soil. The decrease in the water level increases as the well spacing decreases or the permeability coefficient increases. When the soft soil permeability coefficient is $1 \times 10^{-8}$ m/s and the well spacing is 2 m, the decrease in the water level can reach 9.72 m after 100 days of drainage.

(3) Both Jacob's model and the numerical model in this paper assume that the water level in the drainage well is fixed. However, the Jacob model considers single-well drainage, and the numerical model in this paper considers multi-well drainage. The two models basically are identical at the initial stage of drainage, after which the deviation begins to appear. The difference increases gradually with time.

Siphon drainage is a new drainage treatment technology, and there are still many issues that need in-depth study. This paper mainly studies the decrease in the groundwater level, but has not conducted in-depth research on the settlement and deformation of soil. When the groundwater level decreases, other methods can be combined to accelerate soil consolidation. The combination of siphon drainage and other methods lacks corresponding research.

**Author Contributions:** Software, validation, writing—original draft, Q.S.; resources, C.W.; investigation, J.W.; data curation, S.Y.; methodology, Y.S.; writing—review & editing, H.S. All authors have read and agreed to the published version of the manuscript.

**Funding:** This research received no external funding.

**Data Availability Statement:** Not applicable.

**Acknowledgments:** The authors thank the Zhejiang Seaport Investment and Operation Group Co., Ltd., Zhoushan, China, for providing the project background of this study.

**Conflicts of Interest:** The authors declare no conflict of interest.

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
