# Peer review of "Calculation Model of Multi-Well Siphoning and Its Feasibility Analysis of Discharging the Groundwater in Soft Soil"

_water, doi:10.3390/w15071319_

Round 1

Reviewer 1 Report

The article focuses on calculations of drainage discharge of surface groundwater (dewatering site) using multi-siphon. The numerical calculations are not calibrated but validated with a physical model of an equivalent single (circular). The validation resulted in a large deviation after 100 days of pumping which is two much. I have the following comments:

Please add a detailed flowchart of the methodology.

1- Calibration of the numerical model is essential. Filed experiments (based on the multi-siphon) should be conducted, or data collected from field experiments should be used. Once the author finishes this task, the presented validation can be accepted. 

If a field experiment is not possible, a lab experiment can be a substitute.

2- What is meant by "good agreement". It is a fuzzy term. Accuracy should be assessed quantitatively. Then, you can decide whether it is good or not.

3- How the results are good and the deviation is about 28% after 100 days?

Reviewer 2 Report

This manuscript aims to present a numerical solution and results of the two-dimensional axisymmetric spatial equation for groundwater seepage.

In the Abstract (line 15) "...the new model..." is mentioned and repeated several times throughout the manuscript. However, the mathematical model is classic and the numerical model is primary (and equally classic) and is not the most appropriate for current applications in the area. So I don't think it's a new model. Elaborate on that.

Lines 71-72 read "As a result of the interaction between the various drainage wells, there is a large difference between multi-well drainage and single-well drainage." and lines 81-83 read "This paper presents a new equivalent calculation model based on the theory of unsteady-well flow and the characteristics of multi-well siphoning." Lines 155-156 read "the problem of multi-well siphoning is transformed into the problem of single-wells." Consistency and additional clarifications are required.

Intending to be a sufficiently robust model for multi-well siphoning applications, its validation is carried out in a very simplified physical model test. Thus, in addition to the inconsistencies between the objectives and the results shown, the question is: what is new in this manuscript with relevant scientific value?

It appears that the second equation (4) and equation (5) are boundary conditions at any time t, and not just "The initial boundary conditions of the new model..." (line 130). Clarify. 

What about the space and time steps used?

The current content of the Discussion and Conclusions sections is irrelevant.

All figures together with their captions must be self-explanatory. Not all of them fulfill this request, especially the cases in Figures 2, 4, 5, 7, 8 and 9.

Citations and references (List of References) don't comply with the journal guidelines. Authors are advised to comply with the journal submission guidelines. 

The English language is moderate. Check all parts of the manuscript and correct grammatical constructions. The authors should ask the help of a native English-speaking proofreader because there are some linguistic mistakes that should be fixed.

Round 2

Reviewer 1 Report

The article is largely improved. 

Author Response

We are very grateful to your comments for the manuscript.

Your suggestions are valuable to improve this manuscript.

Reviewer 2 Report

The authors have done a good job addressing the questions raised by reviewers in the revised submission. My most relevant concerns have been satisfactorily addressed.

Essentially, a new figure, a field test, and a new reference have been added, along with English improvements. Unnecessary content in the Discussion section has also been removed.

Anyway, I have two major comments and I think they are critical to be addressed by the authors prior to publication; they are:  

- It would be desirable to provide some details about the numerical model. Since this is an explicit model, questions are raised about the space and time steps that guarantee the best numerical solution. It would be desirable to show a comparison of numerical results with Jacob's analytical solution for the same validity conditions.

- The Conclusion section focuses on the essential points addressed in the manuscript. However, it should start with a paragraph framing the work done. In addition, it should also include a last paragraph with recommendations for future research.

I encourage authors to address these two items as I believe this is of interest to the community.

Other minor points are:

- Avoid using keywords that already exist in the title, such as "soft soil".

- Check the captions of Figures 12 and 13; it seems that b) is switched in both captions.

- Authors are advised to follow journal submission guidelines, especially citations (Wu et al., line 94, and numbering should not be in superscript) and references (some years in bold).
